# Gut Microbiota Modulation in the Context of Immune-Related Aspects of *Lactobacillus* spp. and *Bifidobacterium* spp. in Gastrointestinal Cancers

**DOI:** 10.3390/nu13082674

**Published:** 2021-07-31

**Authors:** Karolina Kaźmierczak-Siedlecka, Giandomenico Roviello, Martina Catalano, Karol Polom

**Affiliations:** 1Department of Surgical Oncology, Medical University of Gdansk, 80-210 Gdańsk, Poland; surgoncolclub@gmail.com; 2Department of Health Sciences, Section of Clinical Pharmacology and Oncology, University of Florence, Viale Pieraccini, 6, 50139 Florence, Italy; giandomenico.roviello@unifi.it (G.R.); marti_cat@yahoo.it (M.C.)

**Keywords:** gut microbiota, immune system, gastrointestinal cancers, immnotherapy

## Abstract

Accumulating evidence has revealed the critical roles of commensal microbes in cancer progression and recently several investigators have evaluated the therapeutic effectiveness of targeting the microbiota. This gut microbiota-related approach is especially attractive in the treatment of gastrointestinal cancers. Probiotics supplementation is a microbiota-targeted strategy that appears to improve treatment efficacy; *Lactobacillus* spp. and *Bifidobacterium* spp. are among the most commonly used probiotic agents. These bacteria seem to exert immunomodulatory effects, impacting on the immune system both locally and systemically. The gut microbiota are able to affect the efficiency of immunotherapy, mainly acting as inhibitors at immune checkpoints. The effects of immunotherapy may be modulated using traditional probiotic strains and/or next generation probiotics, such as *Akkermansia municiphila*. It is possible that probiotics might enhance the efficiency of immunotherapy based on PD-1/PD-L1 and CTLA-4 but more data are needed to confirm this speculation. Indeed, although there is experimental evidence for the efficacy of several strains, the health-promoting effects of numerous probiotics have not been demonstrated in human patients and furthermore the potential risks of these products, particularly in oncologic patients, are rarely mentioned.

## 1. Introduction

The gut microbiota is described as a complex ecosystem, which includes bacteria, viruses, fungi, protozoa, and *Archeae* [1,2] that interact with each other and with the host. These interactions affect the host’s physiopathology and are involved in maintaining homeostasis [2]. The gut microbiota has important roles in the human body e.g., its interaction with gut immunity, its ability to regulate the level of secondary bile acids, its influence on metabolites produced in the gut [1,3]. Therefore, a gut microbiota imbalance may significantly contribute to the development of multiple local and systemic diseases, including gastrointestinal cancers. Thus, an appropriate modulation of gut microbiota could be useful in preventing the development and progression of gastrointestinal cancers and also may be beneficial in supporting effective treatments.

There are available several therapeutic methods being used to modify the composition of the gut microbiota, such as administration of prebiotics, probiotics, synbiotics as well as postbiotics, and faecal microbiota transplantation (FMT) [1]. Currently, probiotics are the most commonly used agents to modify gut microbiota in multiple conditions.

The word “probiotic” is derived from Greek and it means “for life” [2]. According to the Food and Agriculture Organization of the United Nations (FAO) and the World Health Organization (WHO), probiotics are described as “live microorganisms which when administered in adequate amounts confer a health benefit on the host” [1,4]. *Lactobacillus* spp. and *Bifidobacterium* spp. are two of the most commonly used probiotic agents [2]. Although the probiotic properties of *Lactobacillus* spp. and *Bifidobacterium* spp. have been intensively studied [2,4,5,6,7], their immunomodulatory effects in cancers have not been extensively investigated, especially their impact on the immune system in cases of gastrointestinal cancers is largely unclear. Therefore, after a brief discussion of the gut microbiota imbalance in gastrointestinal cancers, we will discuss the immunomodulatory effects of *Lactobacillus* spp. and *Bifidobacterium* spp. in malignancies. Finally, we will summarize the current knowledge related to the link between gut microbiome and immunotherapy efficacy.

## 2. Gut Microbiota Imbalance in Gastrointestinal Cancer

The gut microbiota is involved in the carcinogenesis process via several species-specific mechanisms, such as triggering inflammation, activation of carcinogens as well as tumorigenic pathways, and damaging host DNA [8]. The association of gut microbiota with gastrointestinal carcinogenesis has been investigated, mainly due to recent advances in sequencing technology. There are known to be symbiotic interactions between resident micro-organisms and the digestive tract contributing to the maintenance of gut homeostasis. However, alterations to the microbiome caused by environmental changes (e.g., infection, diet and/or lifestyle) can disturb this symbiotic relationship and promote disease, such as inflammatory bowel disorders and cancer. Indeed, a shift in microbiota profile is claimed to be associated with the development and progression of gastrointestinal cancer [9].

Certain microbe-associated molecular patterns (e.g., flagellins, lipopolysaccharides) can be identified by recognition receptors of the innate immune system and trigger an enhanced toll-like receptor-mediated immune response leading to a persistent inflammation that can worsen further the imbalanced microbial community, thus forming a vicious cycle, eventually resulting in the appearance of gastrointestinal carcinogenesis [10,11,12,13]. A gut microbiota imbalance in patients with gastrointestinal cancer can be caused by multiple factors such as the adverse effects of chemotherapy and infectious agents (Figure 1) [14]. The gut microbiota imbalance seems to be cancer type-specific, characterized by an increase of specific bacteria strains in different types of gastrointestinal cancers (Table 1). Notably, different bacterial as well as fungal species are involved in carcinogenesis in particular via distinctive species-specific mechanisms [15,16]. For instance, *Fusobacterium nucleatum* modulates the E-cadherin signaling pathway as well as activating T-cell factor, b-catenin, NF-kB, c-myc, and cyclin D1. Consequently, it enhances the proliferation of colon cancer cells. Another bacterial species, i.e., *Helicobacter hepaticus* takes part in carcinogenesis through the stimulation of pro-inflammatory mediators (IL-1β, IL-6, IL-8, IFN-γ, TNF-α) production. *Helicobacter pylori* activates NF-kB and activator protein-1 (AP-1) leading to a dysregulation of cellular processes. Additionally, *H. pylori* increases the expression of Bcl-xL, MCL-1, survivin, c-myc, and cyclin D-1. Moreover, pathogenic components, such as ammonia and lipopolysaccharide derived from *H. pylori*, contribute to pancreatic damage [15]. Notably, some bacterial metabolites may enter the bloodstream and consequently they can alter the systemic immune system [17].

As was mentioned above, not only are bacterial microbiota involved in carcinogenesis, but there is also a fungal contribution. Similarly to bacteria, changes evoked by fungi are also species specific-dependent. For instance, *Candida*—fungal genus —produces carcinogenic byproducts, triggers inflammation, and increases the proliferation and activation of myeloid-derived suppressor cells (MDSCs) [18,19]. Another fungal genus, *Trichosporon*, has been reported to increase the level of proinflammatory mediators, such as IL-6, IFN-γ, TNF-α, and granulocyte-colony-stimulating factors (G-CSF) [20,21].

The immune-cell composition of the tumour microenvironment may be altered by specific species, for example, *Peptostreptococcus anaerobius* and the enterotoxigenic *Bacteroides fragilis* (ETBF) trigger chemokine secretion by recruiting immunosuppressive MDSCs, tumour-associated macrophages, and tumour-associated neutrophils [22].

To conclude, there are multiple gut microbiota-dependent mechanisms involved in carcinogenesis. Microbes are known for their pathogenicity and carcinogenicity. Different gut dysbiosis may occur in particular type of gastrointestinal cancers.

## 3. The Probiotics and the Interactions with Immune System

After over a century of investigation, *Bifidobacterium* (*adolescentis*, *animalis*, *bifidum*, *breve*, and *longum*) and *Lactobacillus* (*acidophilus*, *casei*, *fermentum*, *gasseri*, *johnsonii*, *paracasei*, *plantarum*, *rhamnosus*, and *salivarius*) have become the most widely used species as probiotics [3]. The main criteria for probiotic strains are safety, functionality, and technological useability [28]. The properties of probiotics are species-dependent; nevertheless, their general actions, as well as those in the prevention of cancer development, may be listed as follows; binding the carcinogenic compounds, inhibition of pathogens, increasing the levels of antioxidant metabolites, production of anti-tumorigenic compounds, maintaining intestinal barrier integrity, modulation of intestinal immunity [29,30,31].

Additionally, several other strains seem promising, such as *Roseburia* spp., *Akkermansia* spp., and *Faecalibacterium* spp. which are worthy of in-depth investigation [4]. Nevertheless, although probiotics have gained a wide popularity, there are conflicting clinical results for many probiotic strains and formulations and there is still an inadequate understanding about their impact on the host and their interactions with the commensal microbiota.

### 3.1. Lactobacillus spp.

*Lactobacillus* spp. are evidently the most prominent probiotic agents of lactic acid bacteria (LAB) [32]. Notably, commensal *Lactobacillus* species are symbiotic in the human host under physiological conditions [33]. *Lactobacilli* are non-pathogenic bacteria producing many beneficial substances, such as bacteriocins as well as hydrogen peroxide [34]. As mentioned above, there are several bacterial species which can be utilized to modulate the gut microbiota. Herein, we will discuss selected commonly used probiotic species: *Lactobacillus plantarum* 299v (DSM 9843), *L. acidophilus* NCK 2025 as well as *L. casei* BL23 and their bidirectional interactions with the immune system [2].

*L. plantarum* 299v, which was originally isolated from human intestine, is characterized by multiple properties [35,36,37], e.g., it is able to survive in a wide range of pH, demonstrating a high tolerance to acidic conditions in the stomach and the more basic pHs in duodenum [38]. *L. plantarum* 299v also acts against potentially pathogens (e.g., *Enterococcus faecalis*, *Bacillus cereus*, *Yersinia enterocolitica*, *Clostridium difficile*, and *Escherichia coli*) and inhibits their growth in the gut [38]. For instance, *L. plantarum* 299v has been reported to inhibit the adhesion of enteropathogenic *E. coli* to intestinal epithelium due to its ability to stimulate the production as well as the release of mucins (MUCs) [39,40,41]. MUCs are glycoproteins conferring protection in intestinal mucosa surface [39]. Herias et al. have investigated the effect of *L. plantarum* 299v on immune function in gnotobiotic rats [34]. In that study, germ-free rats were divided into 2 groups: one colonized with type 1-fimbriated *E. coli* O6:K13:H1 whereas the other group were administered the same *E. coli* strain combined with *L. plantarum* 299v. It was observed that after 1 week of colonization, rats from the second group displayed lower counts of *E. coli* in small intestine and caecum in comparison to rats from first group. However, at 5 weeks after colonization, the amount of *E. coli* was similar in both groups. Rats from first group had a significantly higher total level of IgA in serum and slightly higher level of IgM as well as IgA antibodies as compared to rats from the second group. Thus, it seemed that *L. plantarum* 299v had increased antibody responses to a gut pathogen, at least in the first 5 weeks. Overall, these results confirmed that *L. plantarum* 299v can directly interfere with *E. coli* colonization [34], and improve the immunological status of the intestinal mucosa. Nevertheless, these results have not yet been confirmed in humans. In the study of Woodcock et al., surgical patients were divided into 2 groups: one receiving a probiotic (*n* = 11) with the other being a control group (*n* = 11) [42]. No significant difference was detected in the concentrations of plasma cells, IgA positive cells or IgM positive cells in lamina propria between groups. However, the control group had a significantly higher concentration of IgM at the gut mucosal surface in comparison to patients receiving probiotics, but the reason was not elucidated [42]. *L. plantarum* 299v can reside on human mucosal cells in vivo [39] and its mechanism of mannose-binding is crucial for its immunomodulating properties. In the study of Rask et al. [43], it was shown that after intake of *L. plantarum* 299v, there were increased expressions of activation markers on CD8+ T cells and a marker for the presence of CD4+ T cells memory cells (CD45RO) [43]. However, as authors suggested, these changes would be more likely to be associated with the action of antigen presenting cells. Nevertheless, it can be speculated that this probiotic strain may improve defence against viral infections.

*L. acidophilus* is able to stimulate the innate cells to produce cytokines via interactions of their surface layer proteins and other cell surface components [33]. In an animal model (generation of TS4Cre × APC^lox^ 468 mice), it was shown that the oral administration of *L. acidophilus* NCK 2025 at a dose 5 × 10^8^ CFU increased the level of anti-inflammatory cytokines, (i.e., IL-10 and IL-12), whereas there was a decline in the level of T regulatory cells (Tregs) [33]. It should be emphasized that Tregs regulate inflammation (through suppression) providing protection against polyposis and the development of colon cancer. Nevertheless, the chronic interaction of Tregs with proinflammatory cells and their cytokines can change their anti-inflammatory properties. Thus, it seems that the regulation of pro- versus anti-tumour immunity is strongly associated with an interaction between lymphocytes and myeloid cells [44,45].

The impact of *L. casei* BL23 on the immune system was studied in an animal model of colorectal cancer (CRC) (female C57BL/6 mice; 6–8 weeks old) [46]. It was found that *L. casei* BL23 downregulated IL-22 providing immunomodulatory effects. Notably, NK cells, Th17, and Th22 cells were the main source of IL-22, but the specific cell-response has been not investigated. Moreover, *L. casei* BL23 was speculated to possess also antiproliferative activities via an upregulation of caspase-7, caspase-9, and Bik, thus increasing cellular apoptosis. Moreover, *L. casei* BL23 reduced histological scores and the value of the proliferative index [46]. Overall, these results indicate a potential role of *L. casei* BL23 in preventing the development of CRC in a mouse model [46].

Recently, in the study of Oh et al., the effect of synbiotic combination (*Lactobacillus gasseri* 505 and *Cudrania tricuspidata* leaf extract) were assessed on colitis-associated colorectal cancer [47]. This synbiotic combination decreased the concentration of pro-inflammatory cytokines (i.e., TNF-α, IFN-γ, IL-1β, and IL-6) as well as enzymes which are related to inflammation, such as inducible nitric oxide synthase (iNOS) and cyclooxygenase-2 (COX-2). Moreover, it up-regulated levels of anti-inflammatory cytokines, i.e., IL-4 and IL-10. Additionally, the levels of biomarkers of mucus layer as well as tight junction aspects (occludin and zonula occludens-1) were up-regulated [47].

### 3.2. Bifidobacterium spp.

The genus *Bifidobacterium* belongs to phylum *Actinobacteria*, which is one of the most abundant phylum in the human gut microbiota (besides *Firmicutes* and *Bacteroidetes*) [48]. *Actinobacteria* is dominated in breast-fed infant whereas in adults one encounters *Firmicutes* and *Bacteroidetes* [48]. Notably, more than 45 species/subspecies belonging to genus *Bifidobacterium,* have been characterized by their high content of guanine as well as cytosine in their genome [49]. *Bifidobacterium* are normal inhabitants of human gastrointestinal tract e.g., *B. adolescentis*, *B. angulatum*, *B. bifidum*, *B. breve*, *B. catenulatum*, *B. dentium*, *B. longum*, *B. pseudocatenulatum*, and *B. pseudolongum* are commonly found in the human gut [49].

Zhang et al. investigated the effect of viable *Bifidobacterium* supplementation administered orally on the composition of the gut microbiota, the properties of the immune system, and prognosis of patients undergoing resection due to colorectal cancer [50]. This study included 60 patients randomly divided into 2 groups: the first (*n* = 30, treatment group) receiving enteral nutrition and orally viable *Bifidobacterium* supplementation before surgery and the second group (*n* = 30, controls) receiving only enteral nutrition. Preoperative and postoperative *Bifidobacterium*/*E. coli* ratios in the control group were significantly lower than in the treatment group (0.72 +/− 0.14, 0.02 +/− 0.06; *p* < 0.05). On day 9 after the operation, in the treatment group there were higher levels of stool sIgA (secretory immunoglobulin A), with anti-inflammatory properties on the mucosal surface mediated by mucosal dendritic cells [51].

Zhang et al. also noted that the serum concentrations of IgG, IgM, IgA, IL-6, and CRP were lower in the treatment group (*p* < 0.05). Moreover, postoperative septic complications were less commonly observed in the treatment group as compared to control group. However, other complications and the duration of hospitalization were similar. Overall, the administration of viable *Bifidobacterium* supplement before surgery for colorectal cancer may alter the composition of the gut microbiota, helping to restore its balance and it may also improve intestinal immunity as well as reducing postoperative complications [50].

Finally, the immunomodulatory effects of *B. longum* KACC 91563 in mouse splenocytes and macrophages was examined by Choi et al. [52]. It was noted that this strain could regulate the proliferation of T and B cells. Moreover, it inhibited the balance between Th1/Th2 cytokines (i.e., Th1: IL-2, TNF-α and Th2: IL-4, IL-10). Additionally, after the administration of *B. longum* KACC 91563, the IgE level was elevated. Thus, this strain seems to be able to modulate the hosts’ immune system via IgE production as well as acting via the maintenance and improvement in the Th1/Th2 balance [52].

The summary of the main properties of *Lactobacillus* spp. and *Bifidobacterium* spp. is included in Figure 2.

## 4. The Link between Immunotherapy and Gut Microbiome

The gut microbiome is able to modulate the host’s immune response locally as well as systemically [53]. Various studies have demonstrated that the composition of the gut microbiota can influence the efficacy of immunotherapy, mainly during treatment with immune checkpoints inhibitors (ICIs) [1,53]. The aims of immune checkpoint blockade are to restore and strengthen the anticancer response by suppressing the intrinsic immuno-inhibitory pathways; these are commonly utilized by tumour cells to develop immune resistance. Much efforts has been invested to exploit the efficacy of treating cancer patients with fully-humanized monoclonal antibodies against two of the most widely studied immune checkpoint regulators—cytotoxic T lymphocyte-associated antigen-4 (CTLA-4) and programmed cell death protein-1 (PD-1) or its ligand PD-1-ligand 1 (PD-L1).

Currently, several ICIs including blockers of PD-1 (nivolumab, pembrolizumab, and cemiplimab), PD-L1 (atezolizumab, avelumab, and durvalumab), and CTLA-4 (ipilimumab) have received approval by the US Food and Drugs Administration (FDA) for treating cancers, having been shown to improve overall survival (OS) of cancer patients [54,55]. Nevertheless, ICIs are associated with adverse events related to the immune system, such as diarrhoea, colitis, dermatologic events as well as liver and lung disorders [55,56,57]. Therefore, there is a strong need to find new therapeutic approaches that could improve the efficacy and safety of immunotherapy. It should be emphasized that gut microbiota could have a significant role in the development of mucosal immune system and intestinal immune homeostasis [55]. Thus, an appropriate modulation of gut microbiota may represent a new therapeutic option. The effects of immunotherapy may be improved by modification of gut microbiota through the administration of probiotics or performing FMT [1,54]. Bacterial strains can exert a significant impact on immunotherapy via enhancement of PD-1/PD-L1 blockade as well as via the stimulation of T cells and CTLA-4 blockade [58,59]. The significant contribution of different commensals in the positive response to immunotherapy treatment against different types of cancer has been revealed in several studies. Next generation sequencing has been exploited to investigate the correlation between gut microbiota and the therapeutic response in patients treated with PD-1/PDL-1 blockade by comparing the diversity and composition of faecal microbiota in responders (R) with non-responders (NR), have been utilized. Although no specific species of bacteria have been identified to be required for PD-1 blockade efficacy, the presence of bacteria of the genus *Bifidobacterium* has been strongly associated with an appropriate response to anti-PD-1 therapy [60]. In addition, high concentrations of *Akkermansia muciniphila* and *Ruminococcacaea* spp. in the gut microbiota have been associated with favourable responses to anti-PD-1 therapy.

An increased abundance of other commensals such as *Enterococcus faecium*, *Collinsella aerofaciens*, *Bifidobacterium adolescentis/longum*, *Klebsiella pneumoniae*, *Veillonella parvula*, *Parabacteroides merdae*, and *Lactobacillus* spp. was also observed in responders by Matson et al. [61] whereas other species such as Roseburia and Faecalibacterium spp. were identified by Maia et al. to be increased in anti-PD-1 responders.

Routy et al. observed that the abundance of *Akkermansia muciniphila* positively affected the clinical responses to ICIs [62]. They demonstrated that oral administration of this strain to NR restored the efficiency of PD-1 blockade in an interleukin-12-dependent manner by increasing the recruitment of CCR9+CXCR3+CD4+ T lymphocytes [62]. Notably, *A. muciniphila* is an anaerobic bacterium and it belongs to next generation probiotics (NGPs) involved in mucous catabolism and has been detected in healthy individuals without disease [62].

A recolonization of ATB-treated mice raised under specific pathogen free (SPF) conditions (or alternatively germ-free animals) by FMT was performed using patient stool samples of feces harvested at diagnosis from R and NR NSCLC patients in an experiment assessing whether there was a correlation between *A. muciniphila* and the response to PD-1/PDL-1 inhibitors. This in vivo test corroborated the clinical data i.e., mice receiving FMT from responders and therefore exhibiting a marked presence of *A. muciniphila*, demonstrated a better response to immuno-oncological therapies and a significant reduction in tumour size with a greater accumulation of immune cells at the level of the cancerous microenvironment [62].

Sivan et al. also indicated that alterations of the gut microbiome could modulate cancer immunotherapy [60]. A slower tumour growth and beneficial responses to anti-PD-1 therapy were observed in mice with a significantly increased amount of *Bifidobacterium* species [53,60]. The oral administration of probiotics containing *Bifidobacterium* increased the anti-tumour efficiency of PD-L1 blockade. Moreover, the presence of *Bifidobacterium* was shown to be positively associated with antitumor T cell responses due to an improvement of tumour-specific CD8^+^ T cell activity, indicating that certain species of this genus, such as *Bifidobacterium breve*, *Bifidobacterium longum and Bifidobacterium adolescentis*, can elicit beneficial antitumor immune effects [60]. Mager et al. revealed [63] that *Bifidobacterium pseudolongum*, *Lactobacillus johnsonii*, and *Olsenella* species were able to significantly improve the efficacy of ICIs in mouse models of cancer. One of these bacterial species, i.e., *B. pseudolongum*, was reported to enhance this therapy via the production of the metabolite inosine. Thus the microbiome-derived inosine was claimed to modulate the response to checkpoint inhibitor immunotherapy [63].

The specific mechanism by which *Bifidobacteria* or other commensal bacteria stimulate antitumor immune responses remains to be elucidated. However, it has been shown that these bacteria can stimulate the maturation of dendritic cells (DC) with a subsequent IL-2 production by the DCs present in the lamina propria of the GI tract. DCs, like antigen-presenting cells (APC), process and present antigens and thus play a role in activating T-cells. CTLA-4 is a homolog of the APC receptor that binds with higher affinity and downregulates T-cell activation. Anti-CTLA-4 monoclonal antibodies block this interaction, a phenomenon favouring T-cell activation and proliferation [64].

Vétizou et al. observed that anti-CTLA-4 therapy treatment significantly altered the gut microbiome of mice by increasing the proportion of *Clostridiales* as well as decreasing those of *Bacteroidales* and *Burkholderiales* [65]. Moreover, mice treated with broad-spectrum antibiotics or GF mice that lack some bacterial species, in particular *Bacteroides*, were resistant to CTLA-4 blockade therapy.

Notably, after the oral administration of bacteria (*Bacteroides fragilis* with *Bacteroides thetaiotaomicron* or *Burkholderia*), the efficacy of anti-CTLA-4 therapy was enhanced by the triggering of a Th1 response and by promoting DC maturation [65]. A similar significant response was observed in cases of fecal transplantation of *Bacteroides* species in GF mice. This suggested that a microbiota-dependent activation of T cells was required to achieve the response to anti-CTLA-4 antibodies [65]. In addition, it has been stated that those patients with melanoma that had microbiomes enriched in the *Faecalibacterium* genus and other *Firmicutes* exhibited longer progression-free survival (PFS) as well as OS when treated with anti-CTLA-4 therapy [66].

Overall, the manipulation of gut microbiota is able to regulate the host’s immune response and may contribute to an improvement of the efficiency of immunotherapy [67]. It also should be noted that blockades of CTLA-4 and PD-1/PD-L1 alter the composition of gut microbiota, causing an injury of intestinal epithelial cells and consequently lead to a loss of intestinal barrier integrity [55].

Currently, several trials regarding gut microbiome-related aspects and immunotherapy in cancer are ongoing worldwide. We surveyed data included in ClinicalTrials.gov (accessed on 12 April 2021) system and several records were found (till July 2021) with one of them apparently being withdrawn (Table 2). These projects are concerned in treating patients with different types of gastrointestinal cancers, such as pancreatic, colorectal, and gastric cancers. The project with the identifier NCT04638751 is focusing not only pancreatic and colorectal cancer patients, but also it has recruited participants with non-small cell lung cancer and triple negative breast cancer. The estimated completion dates of these trials vary (NCT03891979—withdrawn, NCT02960282—April 2023, NCT04744649—December 2024, NCT04638751—December 2024).

The gut microbiota has an impact on the efficiency of anti-cancer therapy not only regarding immunotherapy, but also surgical treatment, chemotherapy, and radiotherapy [15]. The development of colorectal anastomotic leakage may be related to a low microbial diversity and higher amounts of mucin-degrading members of the *Bacteroidaceae* and *Lachnospiraceae* families [68]. Shogan et al. revealed that *Enterococcus faecalis* could contribute to the development of an intestinal anastomotic leak via its collagen-degrading properties as well as its ability to activate matrix metalloproteinase 9 (MMP9) [69]. *Fusobacterium nucleatum* was reported to promote resistance to chemotherapy through activation of autophagy. *F. nucleatum* targeted TLR4 and MYD88 innate immune signalling and specific microRNAs [70]. Additionally, *F. nucleatum* up-regulated the expression of BIRC3 in colorectal cancer and consequently promoted chemoresistance to 5-fluorouracil [71]. The activity of gut microbiota has also an impact on the efficiency of adjuvant 5-fluorouracil chemotherapy and the occurrence of gastrointestinal symptoms related to this anti-cancer treatment [72]. Holma et al. demonstrated that patients who were colonic methane producers had less frequent diarrhoea than non-producers [odds ratio (OR), 0.42; 95% confidence interval (CI), 0.20 to 0.88; *p* = 0.022] [72].

The most frequent side effects of pelvic radiotherapy are fatigue and diarrhoea. In a pilot study, it was noted that signs of gut microbiota dysbiosis may predict these symptoms in patients undergoing pelvic cancer radiotherapy. Especially, the presence of a high microbial diversity prior to radiotherapy may be crucial in this context [73]. Probiotics have been able to prevent the development of radiation-induced diarrhoea. In the study of Delia et al., patients were divided into two groups: the first receiving VSL#3 (*n* = 239) and the second administered placebo (*n* = 243) [74]. Patients who received probiotics were less likely to suffer from radiation-induced diarrhoea (31.6% vs. 51.8%; *p* < 0.001, respectively) as well as 3 and 4 grade diarrhoea compared to placebo recipients (55.4% vs. 1.4%, *p* < 0.001) [74].

Moreover, the gut microbiota may be used as a prognostic biomarker linked with the survival of colorectal cancer patients, as was shown in the pilot study conducted by Wei et al. [75]. In that work, the gut microbiota was assessed using 16S rRNA gene sequencing; a higher abundance of *Faecalibacterium prausnitzii* was noted in the surviving group, whereas higher amounts of *F. nucleatum* and *Bacteroides fragilis* were observed in patients with a poorer prognosis [75].

## 5. Conclusions

*Lactobacillus* spp. and *Bifidobacterium* spp. have been the most widely studied and used probiotic agents but their immunomodulatory properties have been relatively poorly evaluated. Nevertheless, there are published data confirming the ability of these probiotics to influence the immune system both locally and systemically. Additionally, the growing interest in unravelling the link between the gut microbiome and immunotherapy may significantly contribute to improvement of efficiency and safety of certain anti-cancer treatments. Therapeutic modification of the gut microbiome is speculated to contribute to regulating the host’s immune response. Administration of particular probiotic strains (for instance *Bifidobacterium*) or NGPs, such as *A. municiphila* has been revealed to improve the efficiency of immunotherapy via enhancement of PD-1/PD-L1 or CTLA-4 blockade. However, there are only a few studies investigating the impact of the gut microbiome on modulating the efficacy of immunotherapy. Moreover, many of these results have originated from work conducted on animal models. Further studies should be concentrated on the role of probiotics/NGPs, but also assessments of OS as well as other effects of ICIs in cancer patients.

## Figures and Tables

**Figure 1 nutrients-13-02674-f001:**
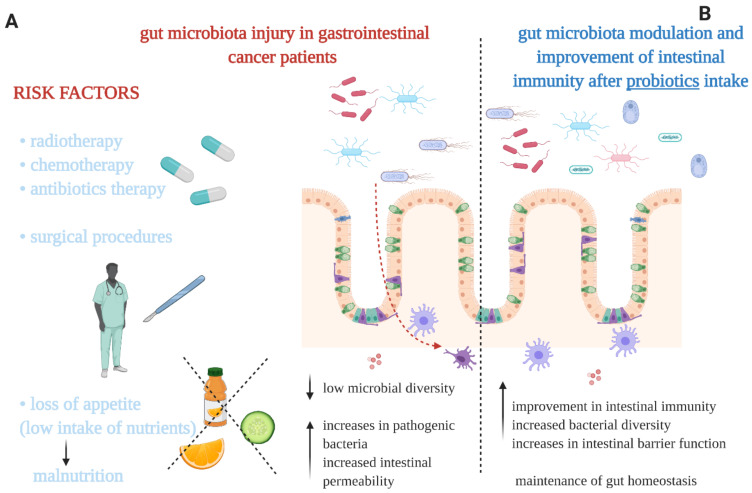
(**A**) The main factors causing a gut microbiota imbalance in gastrointestinal cancer patients. (**B**) Some of the potential effects on gut microbiota and intestinal immunity evident after administration of probiotics. Our proposals based on the literature [1,14,15,23].

**Figure 2 nutrients-13-02674-f002:**
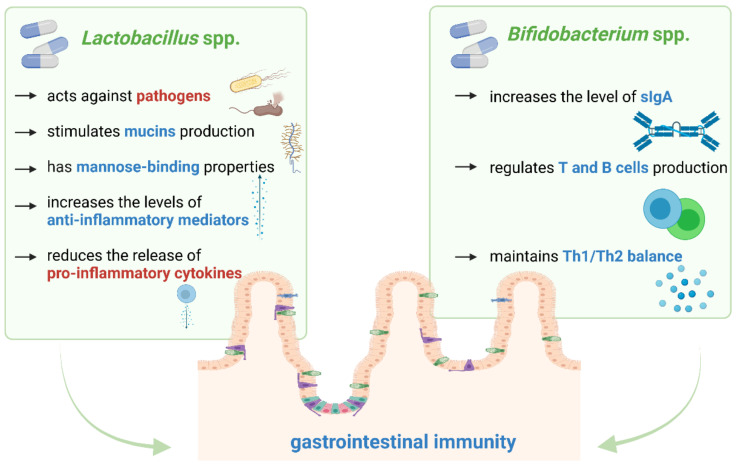
The main properties of *Lactobacillus* spp. and *Bifidobacterium* spp. through which they modulate gastrointestinal immunity. Our proposals based on literature [38,39,46,50,51,52].

**Table 1 nutrients-13-02674-t001:** Bacterial gut microbiota imbalance in selected gastrointestinal cancers.

Gastrointestinal Cancers	Bacteria Altered Composition
Gastric cancer	*Helicobacter pylori* *Lactococcus* *Veilonella* *Fusobacteriaceae* [22]
Colorectal cancer	*Enterococcus faecalis* *Fusobacterium nucleatum* *Streptococcus bovis* *Escherichia coli* *Peptostreptococcus anaerobius* *Bacteroides fragilis* *Helicobacter hepaticus* *Porhyromonas gingivalis* *Helicobacter pylori* *Streptococcus gallolyticus* *Clostridium septicum* [10,15,24,25]
Pancreatic cancer	**Oral microbiota:** *Porphyromonas gingivalis* *Fusobacterium* *Neisseira elongata* *Streptococcus mitis* *Bacteroides* *Lepotrichia* *Grabulitacetlla adiacens* *Aggregatibacter actinomycetemocomitans* **Intrapancreatic microbiota:** *Gammaproteobacteria* *Fusobacterium* *Escherichia coli* *Bifidobacterium pseudolongum* **Bactibilia:** *Enterococcus faecalis* *Escherichia coli* *Helicobacter pylori* infection [25,26]
Liver cancer	*Veillonella* *Streptococcus* [27]

**Table 2 nutrients-13-02674-t002:** The current project registered in ClinicalTrials.gov system regarding gut microbiome-related aspects and immunotherapy in cancer.

ClinicalTrials.gov Identifier	Title of Project	Study Type	Disease/Condition	Estimated Enrollment of Participants (*n*)	Intervention/Treatment	Current Status
NCT03891979	“Gut microbiome modulation to enable efficacy of checkpoint-based immunotherapy in pancreatic adenocarcinoma”.	Pilot study	Pancreatic cancer	No available data	Drug: Pembrolizumab Drug: Ciprofloxacin 500 mg PO BID days 1–29 Drug: Metronidazole 500 mg PO TID days 1–29	Withdrawn
NCT02960282	“Gut microbiome in fecal samples from patients with metastatic cancer undergoing chemotherapy or immunotherapy”.	Observational study	Metastatic carcinoma; stage IV colorectal cancer	80	Procedure: Biospecimen Collection Other: Laboratory Biomarker Analysis	Recruiting
NCT04744649	“Neoadjuvant immunotherapy and chemotherapy for locally advanced esophagogastric junction and gastric cancer trial”	Interventional study	Gastric cancer	80	Drug: XELOX or SOX XELOX: Oxaliplatin+Capecitabine; SOX: Oxaliplatin+S-1	Recruiting
NCT04638751	“ARGONAUT: stool and blood sample bank for cancer patients”.	Observational	Non-small cell lung cancer; triple negative breast cancer; colorectal cancer and pancreatic cancer.	4000	Drug: Immunotherapy Drug: Chemotherapeutic Agent	Recruiting

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
