# Peer review of "Gut Microbiota Modulation in the Context of Immune-Related Aspects of Lactobacillus spp. and Bifidobacterium spp. in Gastrointestinal Cancers"

_nutrients, 2021, doi:10.3390/nu13082674_

Round 1
Reviewer 1 Report
This manuscript summarized the current understanding of the role of gut microbiota modulation in the context of immune-related aspects of probiotics, mainly Lactobacillus spp. and Bifidobacterium spp., in GI cancers pathogenesis and immunotherapy. This topic is needed in the area considering the emerging evidence showing that microbiota dysbiosis plays a role in cancer development and therapy, and that probiotics have immunomodulatory effects on GI cancers. However, this review was not very well organized and comprehensively described, with some parts not relevant to the topic. It will be great if the authors can cover more details on the topic, i.e., microbiome and probiotics in GI cancers, and include adequate references to related and previous work. Several major concerns are list as below.
Comments:
- Section 2 microbiota in GI cancers: The role of microbiome in GI cancers is more complicate than what the authors described here, including Figure 1 and Table 1 (ref#10, 15-18). Many important previous related papers were missing, to name a few: Gopalakrishnan, V., et al. (2018). "The Influence of the Gut Microbiome on Cancer, Immunity, and Cancer Immunotherapy." Cancer Cell 33(4): 570-580. Janney, A., et al. (2020). "Host-microbiota maladaptation in colorectal cancer." Nature 585(7826): 509-517. LaCourse, K. D., et al. (2021). "The relationship between gastrointestinal cancers and the microbiota." Lancet Gastroenterol Hepatol.
- Section 3 probiotics with immune system: this section mainly described what are probiotics as well as the role of probiotics in infection and immunity; both of which are not relevant to the topic: probiotics in the immunomodulation of GI cancers. Although a few refs (32, 33, 36) discussed here are related to the topic, more work need to be added here, such as: Molska, M. and J. Regula (2019). "Potential Mechanisms of Probiotics Action in the Prevention and Treatment of Colorectal Cancer." Nutrients 11(10). Ambalam, P., et al. (2016). "Probiotics, prebiotics and colorectal cancer prevention." Best Pract Res Clin Gastroenterol 30(1): 119-131. Javanmard, A., et al. (2018). "Probiotics and their role in gastrointestinal cancers prevention and treatment; an overview." Gastroenterol Hepatol Bed Bench 11(4): 284-295. Oh, N. S., et al. (2020). "Cancer-protective effect of a synbiotic combination between Lactobacillus gasseri 505 and a Cudrania tricuspidata leaf extract on colitis-associated colorectal cancer." Gut Microbes 12(1): 1785803. So, S. S., et al. (2017). "Probiotics-mediated suppression of cancer." Curr Opin Oncol 29(1): 62-72.
- Section 4 microbiome and immunotherapy: the majority of the part is about the immunotherapy of non-GI cancers, therefore not quite relevant to the topic. Table 2 of the clinical trials is also a bit off-topic as only 2 trials are for GI cancers. This section should focus on the role of microbiome and probiotics in the therapy of GI cancers.
- There are numerous English language errors throughout. Please thoroughly edit the manuscript for language.
Author Response
Reviewer 1
This manuscript summarized the current understanding of the role of gut microbiota modulation in the context of immune-related aspects of probiotics, mainly Lactobacillus spp. and Bifidobacterium spp., in GI cancers pathogenesis and immunotherapy. This topic is needed in the area considering the emerging evidence showing that microbiota dysbiosis plays a role in cancer development and therapy, and that probiotics have immunomodulatory effects on GI cancers. However, this review was not very well organized and comprehensively described, with some parts not relevant to the topic. It will be great if the authors can cover more details on the topic, i.e., microbiome and probiotics in GI cancers, and include adequate references to related and previous work. Several major concerns are list as below.
Comments:
- Section 2 microbiota in GI cancers: The role of microbiome in GI cancers is more complicate than what the authors described here, including Figure 1 and Table 1 (ref#10, 15-18). Many important previous related papers were missing, to name a few: Gopalakrishnan, V., et al. (2018). "The Influence of the Gut Microbiome on Cancer, Immunity, and Cancer Immunotherapy." Cancer Cell 33(4): 570-580. Janney, A., et al. (2020). "Host-microbiota maladaptation in colorectal cancer." Nature 585(7826): 509-517. LaCourse, K. D., et al. (2021). "The relationship between gastrointestinal cancers and the microbiota." Lancet Gastroenterol Hepatol.
Thank you for all comments which allow us to improve the quality of this manuscript. We totally agree with you that the role of gut microbiome in GI cancers is more complicated than we presented earlier. Thus, according to your comment, we developed this section presenting several mechanisms. We also add articles which were recommended by you.
2. Section 3 probiotics with immune system: this section mainly described what are probiotics as well as the role of probiotics in infection and immunity; both of which are not relevant to the topic: probiotics in the immunomodulation of GI cancers. Although a few refs (32, 33, 36) discussed here are related to the topic, more work need to be added here, such as: Molska, M. and J. Regula (2019). "Potential Mechanisms of Probiotics Action in the Prevention and Treatment of Colorectal Cancer." Nutrients 11(10). Ambalam, P., et al. (2016). "Probiotics, prebiotics and colorectal cancer prevention." Best Pract Res Clin Gastroenterol 30(1): 119-131. Javanmard, A., et al. (2018). "Probiotics and their role in gastrointestinal cancers prevention and treatment; an overview." Gastroenterol Hepatol Bed Bench 11(4): 284-295. Oh, N. S., et al. (2020). "Cancer-protective effect of a synbiotic combination between Lactobacillus gasseri 505 and a Cudrania tricuspidata leaf extract on colitis-associated colorectal cancer." Gut Microbes 12(1): 1785803. So, S. S., et al. (2017). "Probiotics-mediated suppression of cancer." Curr Opin Oncol 29(1): 62-72.
Thank you for the improvement comment. We developed this section and added above proposed as well as other papers.
3. Section 4 microbiome and immunotherapy: the majority of the part is about the immunotherapy of non-GI cancers, therefore not quite relevant to the topic. Table 2 of the clinical trials is also a bit off-topic as only 2 trials are for GI cancers. This section should focus on the role of microbiome and probiotics in the therapy of GI cancers.
According to your comments, we improved section 4. We described the role of gut microbiome and its related aspects in the therapy of GI cancers. We also corrected Table 2 by removing some studies which were not related directly to GI cancers and by adding new studies. We also improved this table through adding an additional data, such as number of participants and intervention/treatment. According to the editor recommend, we also prepared graphical abstract which allow to better understanding of this article.
4. There are numerous English language errors throughout. Please thoroughly edit the manuscript for language.
Thank you for this comment. Whole manuscript was corrected line by line by native speaker.

Reviewer 2 Report
The manuscript “Gut microbiota modulation in the context of immune-related aspects of Lactobacillus spp. And Bifidobacterium spp. In gastrointestinal cancers” by Kaźmierczak-Siedlecka et al. focuses on the immunomodulatory effects of the two probiotic strains, namely Lactobacillus spp. and Bifidobacterium spp.
Altogether, the review addresses an interesting and currently frequently discussed issue, namely the host-microbial interaction in the context of (gastrointestinal) cancers and their contribution to immunotherapy efficacy. However, considering their central concern to discuss the current knowledge on the impact of Lactobacillus spp. and Bifidobacterium spp. as two major probiotics and their potential impact on gastrointestinal malignancies and immunotherapy efficacies, the overall discussion and conclusion on this topic is displayed only insufficiently in this manuscript.
There are a few major points that have to be discussed and explained in more detail:
In Figure 1 the authors summarize the main factors causing gut microbiota imbalance in gastrointestinal cancer patients, yet references are missing. Furthermore, the overall illustration seems misleading, e.g. (B) is a working hypothesis or refers to published studies.
In Table 1 the authors summarize cancer-related bacterial species, however the references are summarized in the table title. The authors should cited the references according to the selected type of gastrointestinal cancer and include the work of Wirbel et al. (Wirbel, J., Pyl, P.T., Kartal, E. et al. Meta-analysis of fecal metagenomes reveals global microbial signatures that are specific for colorectal cancer. Nat Med 25, 679–689 (2019). https://doi.org/10.1038/s41591-019-0406-6).
The authors nicely summarize current knowledge on pro- and anti-inflammatory properties of Lactobacillus spp and Bifidobacterium spp. However, some conclusions are either missing, confusing or simply wrong. An illustration of both bacterial strains and their potential properties on gastrointestinal immunity in health and disease would help for a better understanding on this topic. Considering the section on L.plantarum the authors state that “it was shown that the expression of activation markers on CD8+ T cells was increased after intake of this probiotic” and concluded that L.plantarum “seems to enhance the immune defense against tumor…” This statement seems quite strong in comparison to the described data. In the section on L.casei, the authors cited and summarized a study by Lenoir et al., yet this study was retracted in 2020. Therefore, the authors should exclude this study.
In the last section on ICIs and Bifidobacterium spp. the authors should include the study by Mager et al. (Microbiome-derived inosine modulates response to checkpoint inhibitor immunotherapy; Science; 18 Sep 2020: Vol. 369, Issue 6510, pp. 1481-1489; DOI: 10.1126/science.abc3421)
In Table 2 the authors describe current projects on cancer-immunotherapies and the gut microbiota. A short summary of the overall approach for each project and which ICIs are used should be included.
In general, the authors should check if their description of pro- and anti-inflammatory cytokines are right (e.g. TNF-γ) and carefully reconsider editing of their language and style.
Author Response
Reviewer 2
The manuscript “Gut microbiota modulation in the context of immune-related aspects of Lactobacillus spp. And Bifidobacterium spp. In gastrointestinal cancers” by Kaźmierczak-Siedlecka et al. focuses on the immunomodulatory effects of the two probiotic strains, namely Lactobacillus spp. and Bifidobacterium spp.
Altogether, the review addresses an interesting and currently frequently discussed issue, namely the host-microbial interaction in the context of (gastrointestinal) cancers and their contribution to immunotherapy efficacy. However, considering their central concern to discuss the current knowledge on the impact of Lactobacillus spp. and Bifidobacterium spp. as two major probiotics and their potential impact on gastrointestinal malignancies and immunotherapy efficacies, the overall discussion and conclusion on this topic is displayed only insufficiently in this manuscript.
There are a few major points that have to be discussed and explained in more detail:
In Figure 1 the authors summarize the main factors causing gut microbiota imbalance in gastrointestinal cancer patients, yet references are missing. Furthermore, the overall illustration seems misleading, e.g. (B) is a working hypothesis or refers to published studies.
Thank you for all improvement comments. We totally agree with you that this illustration needs references. Therefore, we add papers on which this figure was based. According to the editor recommendation, we also prepared graphical abstract which allow better understanding of this article.
In Table 1 the authors summarize cancer-related bacterial species, however the references are summarized in the table title. The authors should cited the references according to the selected type of gastrointestinal cancer and include the work of Wirbel et al. (Wirbel, J., Pyl, P.T., Kartal, E. et al. Meta-analysis of fecal metagenomes reveals global microbial signatures that are specific for colorectal cancer. Nat Med 25, 679–689 (2019). https://doi.org/10.1038/s41591-019-0406-6).
Thank you for this significant comment. We corrected Table 1. We also included Wirbel et al. paper.
The authors nicely summarize current knowledge on pro- and anti-inflammatory properties of Lactobacillus spp and Bifidobacterium spp. However, some conclusions are either missing, confusing or simply wrong. An illustration of both bacterial strains and their potential properties on gastrointestinal immunity in health and disease would help for a better understanding on this topic. Considering the section on L.plantarum the authors state that “it was shown that the expression of activation markers on CD8+ T cells was increased after intake of this probiotic” and concluded that L.plantarum “seems to enhance the immune defense against tumor…” This statement seems quite strong in comparison to the described data. In the section on L.casei, the authors cited and summarized a study by Lenoir et al., yet this study was retracted in 2020. Therefore, the authors should exclude this study.
We corrected some conclusions. We also prepared an additional figure presenting the potential properties of these probiotics by which they modulate gastrointestinal immunity. We corrected the section regarding Lactobacillus plantarum 299v. We also excluded Lenoir et al. study.
In the last section on ICIs and Bifidobacterium spp. the authors should include the study by Mager et al. (Microbiome-derived inosine modulates response to checkpoint inhibitor immunotherapy; Science; 18 Sep 2020: Vol. 369, Issue 6510, pp. 1481-1489; DOI: 10.1126/science.abc3421)
According to your comment, we add Mager et al. study in the last section on ICIs and Bifidobacterium spp.
In Table 2 the authors describe current projects on cancer-immunotherapies and the gut microbiota. A short summary of the overall approach for each project and which ICIs are used should be included.
Thank you for this significant comment. We corrected Table 2. We removed studies which were not related directly to the gastrointestinal cancers. We also improved this table adding an additional data, such as number of participants and intervention/treatment.
In general, the authors should check if their description of pro- and anti-inflammatory cytokines are right (e.g. TNF-γ) and carefully reconsider editing of their language and style.
We corrected the description of TNF-α. We checked other name of pro- and anti-inflammatory cytokines. Whole manuscript was also corrected line by line by native speaker.
